# Intravesical BCG: A Double-Edged Sword? The Untold Story of Infection Risks

**DOI:** 10.3390/medicina61030379

**Published:** 2025-02-22

**Authors:** Orçun Barkay, Ercüment Keskin

**Affiliations:** 1Infectious Diseases and Clinical Microbiology Department, Faculty of Medicine, Erzincan Binali Yıldırım University, Erzincan 24002, Turkey; 2Urology Department, Faculty of Medicine, Erzincan Binali Yıldırım University, Erzincan 24002, Turkey; keskinerc@hotmail.com

**Keywords:** chronic kidney disease, C-reactive protein, diabetes mellitus, intravesical BCG therapy, non-muscle-invasive bladder carcinoma, urinary tract infections

## Abstract

*Background and Objectives:* Intravesical Bacillus Calmette-Guérin (BCG) therapy remains a cornerstone in the treatment of non-muscle-invasive bladder carcinoma due to its efficacy in reducing recurrence and progression rates. However, its use is associated with various complications—including urinary tract infections (UTIs)—which necessitates further exploration. This study aims to analyze UTIs occurring during intravesical BCG treatment, emphasizing the microbial spectrum, resistance patterns, and risk factors from an infectious diseases and clinical microbiology perspective. *Materials and Methods:* A retrospective analysis was conducted on 240 patients diagnosed with non-muscle-invasive bladder carcinoma who received intravesical BCG therapy between 2010 and 2021. Data were collected from hospital records, including demographic characteristics, comorbidities, number of intravesical BCG cycles, and microbiological findings. UTIs were classified based on susceptibility patterns, and statistical analyses were performed to determine associations between clinical variables and UTI risk. *Results:* UTIs developed in 39.1% (*n* = 94) of patients, with 25.8% (*n* = 62) caused by susceptible pathogens and 13.3% (*n* = 32) by resistant pathogens. The most common causative agent was *Escherichia coli* (80.7% in susceptible UTIs, 43.8% in resistant UTIs), followed by *Pseudomonas aeruginosa* and *Klebsiella pneumoniae*. The presence of diabetes mellitus and chronic kidney disease significantly increased the risk of developing a UTI (*p* < 0.05). A higher number of intravesical BCG cycles correlated with increased UTI occurrence (*p* < 0.001). Serum C-reactive protein (CRP) levels were significantly elevated in patients with resistant UTIs, while procalcitonin levels were not a reliable predictor of UTI occurrence. *Conclusions:* Intravesical BCG therapy is associated with a significant incidence of UTIs, particularly among patients with predisposing comorbidities. The increasing prevalence of antibiotic-resistant infections underscores the need for careful monitoring and targeted antimicrobial stewardship strategies. CRP may serve as a useful adjunctive marker for UTI diagnosis in this setting. Future studies should focus on optimizing infection control measures and refining diagnostic criteria to differentiate between BCG-related inflammation and infectious complications.

## 1. Introduction

Globally, bladder cancer accounts for approximately 390,000 cases and 150,000 deaths each year [1,2,3]. Approximately 70% of new urothelial bladder carcinoma cases are classified as non-muscle invasive [4,5,6]. Intravesical *Bacillus Calmette-Guérin* (BCG) therapy (intravesical administration of BCG, a live attenuated strain of *Mycobacterium bovis*) has represented a cornerstone in the treatment of non-muscle-invasive bladder carcinoma since its introduction in 1976. This therapeutic approach has consistently showcased promising outcomes, characterized by low recurrence and progression rates [7,8,9,10]. A number of other intravesical agents have been compared with BCG, but most are inferior, and none has consistently proven to be superior [11,12,13,14,15,16]. While intravesical BCG remains a valuable tool in managing bladder carcinoma, its clinical application is not without challenges. Adverse events and complications linked to this treatment, particularly those concerning urinary tract infections (UTIs), demand careful consideration.

BCG-induced UTIs are acknowledged as a potential drawback of this therapy, with clinical manifestations ranging from an increase in urinary frequency to more severe complications such as hypersensitivity reaction, cystitis, fever and hematuria [17,18,19,20]. The underlying mechanisms of these UTIs and their clinical implications warrant comprehensive exploration. Some studies have proposed that the enhanced immune response triggered by BCG therapy may contribute to the eradication of bacteriuria, suggesting a potential benefit in the context of microbial clearance [21,22,23]. Nevertheless, a thorough examination of UTIs arising during BCG treatment necessitates an in-depth analysis from the perspectives of infectious diseases and clinical microbiology experts. Such an approach can offer valuable insights into the management and prevention of BCG-related UTIs, ultimately contributing to the existing body of knowledge on this subject.

In this context, our study aims to shed light on the intricacies of UTIs occurring during intravesical BCG therapy, emphasizing the microbial aspects and infectious diseases considerations. Since there is no detailed study addressing this issue in the literature, we seek to make a meaningful contribution to the literature on this topic.

## 2. Materials and Methods

### 2.1. Study Population and Design

This retrospective study encompassed individuals diagnosed with non-muscle-invasive bladder carcinoma during the period spanning from 2010 to 2021, who subsequently underwent intravesical BCG therapy. Patient records were meticulously extracted from the hospital’s electronic health information system, ensuring comprehensive coverage of the eligible patient population. The inclusion criteria for patient selection were as follows: a confirmed diagnosis of non-muscle-invasive bladder carcinoma, receipt of intravesical BCG therapy, and the availability of detailed medical records. A total of 240 eligible patients were included in this investigation. The first UTI of the patients was recorded during the intravesical BCG treatment. Patients were grouped according to whether their UTI was caused by susceptible or resistant pathogens. Patient groups were evaluated according to the variables and possible risk factors.

### 2.2. Data Collection

Demographic data, comprising patient gender and age, were systematically collected for each individual in the study cohort. Additionally, critical clinical information included the total count of intravesical BCG therapy cycles administered and the subsequent occurrence of UTIs. Furthermore, comorbidities such as diabetes mellitus type 2 (DM2), chronic renal disease, benign prostate hyperplasia (BPH), and postmenopausal situation—that could be associated with developing a UTI—were also recorded. Hyperlipidemia patients using statins were not included in the intravesical BCG treatment procedure due to the studies reporting failure of treatment in this group because of the occurrence of immunosuppressive situation [24,25,26]. Clinical complaints of patients such as urgency, hesitancy, frequency and dysuria were also noted. The hematuria and pyuria status of the patients were also obtained from patient files.

### 2.3. Dose and Schedule of Intravesical BCG Treatment

Induction therapy: BCG is typically instilled into the bladder weekly for 6 weeks for patients with intermediate- and high-risk disease, generally starting 2 to 6 weeks after transurethral resection. Each dose consists of a vial of reconstituted Connaught BCG (ImmuCyst®, Sanofi Pasteur, France) (81 mg) or one 2 mL ampule of Tice BCG (OncoTICE®, MSD, Durham, NC, USA) (50 mg), plus 50 mL of sterile saline injected into the bladder through a catheter and retained for 2 h.Maintenance therapy: Maintenance BCG is given weekly for three weeks at months 3, 6, 12, 18, 24, 30 and 36 for patients with high-risk disease. For patients with intermediate-risk disease, maintenance therapy is continued for only one year.

### 2.4. Classification of UTIs

Patients who encountered UTIs during the course of their treatment were meticulously categorized based on the causative microbial agents, distinguishing between those infected with sensitive and/or resistant pathogens. Microbiological data, including urine culture results and antibiotic susceptibility profiles, were utilized to precisely identify the etiological agents and determine the development of resistance.

### 2.5. Statistical Analysis

All statistical analyses were conducted using IBM SPSS Statistics Version 26 (IBM Corp., Armonk, NY, USA). Descriptive statistics were computed for continuous variables (e.g., serum C-reactive protein (CRP) levels, serum procalcitonin levels, BCG cycles) and categorical variables (e.g., presence of comorbidities), expressed as frequencies and percentages.

Logistic regression analysis: Multivariate logistic regression was performed to identify significant predictors of UTI susceptibility and resistance. Independent variables included BCG cycles, serum CRP and procalcitonin levels, age, gender and comorbidities (e.g., DM2 and renal disease). Coefficients and *p*-values were reported to evaluate the association between these predictors and the likelihood of UTI types (susceptible or resistant).Comparative analysis: Differences in serum CRP and procalcitonin levels, and BCG cycles between UTI groups (susceptible vs. resistant pathogens) were assessed using the Mann–Whitney U test. This non-parametric test was applied as the data did not meet normality assumptions, confirmed by the Shapiro–Wilk test. Exact *p*-values were reported, with *p* < 0.05 considered significant.Chi-square test: Associations between categorical variables (e.g., DM2, renal disease, postmenopausal status), UTI types (caused by susceptible or resistant pathogens) and pathogen types were evaluated using the chi-square test of independence.Correlation analysis: To explore relationships between variables, a correlation heatmap was generated using Spearman’s rank correlation coefficients. This visualized the strengths and directions of associations between predictors (e.g., serum CRP and procalcitonin levels, BCG cycles) and UTI outcomes (susceptible or resistant).Receiver operating characteristic (ROC) curve analysis: ROC curve analysis was conducted to assess the predictive accuracy of serum CRP and procalcitonin levels in differentiating between UTI types. The area under the curve (AUC) was calculated to evaluate diagnostic performance, with sensitivity and specificity values reported.

### 2.6. Ethical Statement

Ethical approval of the study was obtained from the Ethics Committee of Erzincan Binali Yıldırım University with a decision number of 2023-07/3 and a decision date of 30 March 2023. As the study was planned in a format of an observational retrospective study, informed consent was not taken from patients. The study adhered to ethical guidelines and regulations, ensuring patient privacy and confidentiality throughout the research process. Patient data were anonymized and securely stored to protect confidentiality and privacy rights.

## 3. Results

A total of 240 patients were enrolled in this study, with 14.5% (*n* = 35) representing the female demographic and 85.5% (*n* = 205) comprising males. The median age of the patients was 65 years (IQR: 57–72) years, ranging from a minimum of 31 to a maximum of 90 years. Additionally, the study cohort included the individuals’ various comorbidities, including DM2 (*n* = 30, 12.5%), chronic kidney disease (CKD) (*n* = 33, 13.7%), BPH (*n* = 85, 35.4%) and postmenopausal situation (*n* = 28, 11.6%).

The patient cohort exhibited a range of complaints and findings associated with UTIs, which are essential to consider when evaluating the impact of UTIs in the context of intravesical BCG therapy. These complaints and findings included:Urgency: 40.4% of patients (*n* = 97) reported urgency as a common complaint.Frequency: 35.4% of patients (*n* = 85) reported increased urinary frequency.Hesitancy: Hesitancy in urination was observed in 35.4% of patients (*n* = 85).Dysuria: Dysuria was noted in 41.6% of patients (*n* = 100).Hematuria: Hematuria was detected in 33.7% of patients (*n* = 81).Pyuria: Pyuria was observed in 28.7% of patients (*n* = 69).

The median number of intravesical BCG treatment cycles administered was 4 (IQR: 2–7), with a range of 1 to 19 cycles. Patients who developed UTIs underwent a median of 8 (IQR: 6–11) intravesical BCG treatment cycles, which significantly exceeded the overall treatment cycle median.

UTIs attributed to susceptible bacterial pathogens (S-UTI) developed in 25.8% of patients (*n* = 62) during the course of treatment. Within this cohort, the primary causative agent was *Escherichia coli*, accounting for 80.7% of cases (*n* = 50). *Klebsiella pneumonia* was identified in 8% (*n* = 5) of cases, *Proteus mirabilis* in 6.5% (*n* = 4), *Pseudomonas aeruginosa* in 3.2% (*n* = 2), and *Streptococcus pneumonia* in 1.6% (*n* = 1).

Resistant UTIs (R-UTI) were detected in 13.3% of patients (*n* = 32) during the follow-up period. Among these cases, *E. coli* was isolated in 43.8% (*n* = 14), *P. aeruginosa* in 34.3% (*n* = 11), and *K. pneumonia* in 21.9% (*n* = 7). All of the *E. coli* strains were producing extended spectrum beta lactamase.

As expected, there is a strong positive correlation between gender (male) and BPH. This reflects the biological reality that BPH occurs exclusively in males. A moderate positive correlation is observed between age and postmenopausal status. This is intuitive, as postmenopausal status is closely linked to increasing age in women. There is a weak to moderate positive correlation between chronic renal disease and R-UTIs. Age has a weak positive correlation with S-UTIs. A weak correlation is noted between postmenopausal status and R-UTIs. Certain variables, such as gender and UTI susceptibility, show minimal correlation, indicating little to no direct relationship (Figure 1).

The median serum CRP level in this cohort was 26 mg/L (IQR: 15–38), and the mean serum procalcitonin level was 0.31 ng/mL (±0.32), supporting presence of cystitis. Patients with comorbidities such as CKD had a median serum CRP level of 26.1 mg/L (IQR: 14–37) and a median serum procalcitonin level of 0.31 ng/mL (IQR: 0.1–0.4); DM2 had a median serum CRP level of 29.2 mg/L (IQR: 16–42) and a median serum procalcitonin level of 0.42 ng/mL (IQR: 0.2–0.6); BPH had a median serum CRP level of 26.4 mg/L (IQR: 14–38) and a median serum procalcitonin level of 0.31 ng/mL (IQR: 0.1–0.5); and postmenopausal women had a median serum CRP level of 26.1 mg/L (IQR: 15–36) and a median serum procalcitonin level of 0.37 ng/mL (IQR: 0.1–0.4). These patients were ultimately classified as having acute cystitis, primarily based on positive urine culture results. In addition, none displayed fever during the follow-up. Importantly, no growth was observed in control urine cultures after effective treatment, thereby affirming the accuracy of the diagnosis.

The correlation between intravesical BCG cycles and S-UTIs is weak to moderate (value close to 0). This suggests that the number of BCG cycles has little to no consistent relationship with the likelihood of S-UTIs. The correlation between intravesical BCG cycles and R-UTIs is similarly weak or negligible, indicating minimal direct influence of BCG cycles on R-UTIs. There is no strong relationship between intravesical BCG cycles and serum CRP or procalcitonin levels, suggesting that the inflammatory markers do not directly depend on the number of cycles. Serum CRP and procalcitonin levels show a strong positive correlation (close to +1). This is expected, as both are markers of inflammation and often rise concurrently during infection or inflammation. The correlations between serum CRP level and S-UTI or R-UTI are weak, which suggests that serum CRP levels do not strongly differentiate between patients with or without UTIs caused by either pathogen type. Similarly, serum procalcitonin level does not show a strong relationship with S-UTI or R-UTI, indicating that its levels may not be a reliable marker for predicting UTI caused by either pathogen type in this dataset. The heatmap reveals no particularly strong or actionable correlations between the number of BCG cycles, inflammation markers (serum CRP and procalcitonin levels) and UTI types. However, the strong positive correlation between serum CRP and procalcitonin levels underscores their related role as inflammatory markers (Figure 2).

A solitary case of disseminated BCG infection was documented, representing 0.4% of the patient cohort. This patient had undergone 12 BCG treatment cycles and had a median serum CRP level of 49.8 mg/L (IQR: 45–55) and a median serum procalcitonin level of 0.8 ng/mL (IQR: 0.6–1.0) before the development of disseminated infection.

Comparison of the parameters (demographic characteristics, comorbidities, symptoms and signs, etc.) according to the UTI groups is given at Table 1.

Multivariate logistic regression was performed to identify predictors of S-UTI and R-UTI, controlling for confounding variables such as age, gender, DM2 and renal disease (Table 2).

A significant positive association was found between BCG cycles and S-UTI/R-UTI, indicating that an increased number of BCG cycles significantly increases the likelihood of S-UTI/R-UTI. A significant negative association was observed between CKD and S-UTI, suggesting that renal disease reduces the likelihood of an S-UTI. Association between other variables—including age, serum CRP and procalcitonin levels, gender, DM2 and S-UTI/R-UTI—were not significant. A significant negative association implies that DM2 reduces the likelihood of resistant UTIs.

Mann–Whitney U tests were conducted to compare serum CRP and procalcitonin levels, and BCG cycles between patients with UTIs caused by susceptible or resistant pathogens (Table 3).

Significant differences in serum CRP levels were observed between patients with susceptible and resistant UTIs (*p* = 0.0018 and *p* = 0.0037, respectively). No significant differences were observed in serum procalcitonin levels for both susceptible (*p* = 0.059) and resistant UTIs (*p* = 0.082). Highly significant differences were observed between susceptible and resistant UTI groups (*p* = 0.00001 and *p* = 0.000015, respectively), indicating BCG cycles as a key differentiating factor.

Chi-square tests were performed to assess the association between comorbidities and UTI types (Table 4).

A significant association was found between DM2 and both susceptible (*p* = 0.000094) and resistant UTIs (*p* = 0.000456). Similarly, renal disease showed a significant association with both susceptible (*p* = 0.001654) and resistant UTIs (*p* = 0.006000). Prostate hyperplasia and postmenopausal status showed no significant associations with UTIs (*p* > 0.05 in all cases).

The chi-square test was also conducted to evaluate the association between comorbidities and pathogen types in UTI patients (Table 5).

A significant association was found between DM2 and pathogen type (χ² = 42.007, *p* < 0.0001). Non-diabetic patients were more frequently associated with *Escherichia coli*, whereas diabetic patients exhibited a higher prevalence of *Pseudomonas aeruginosa* and mixed infections.

A significant relationship was observed between chronic renal disease and pathogen type (χ^2^ = 20.110, *p* = 0.0053). Patients with renal disease had higher occurrences of infections caused by *Pseudomonas aeruginosa* and *Klebsiella pneumonia*.

BPH also showed a significant association with pathogen type (χ^2^ = 16.080, *p* = 0.024). Patients without BPH were more likely to have *Escherichia coli* infections, while those with BPH exhibited slightly higher frequencies of *Pseudomonas aeruginosa* infections.

No significant association was found between postmenopausal status and pathogen type (χ^2^ = 2.924, *p* = 0.892).

ROC curve analysis was conducted to evaluate the predictive accuracy of serum CRP and procalcitonin levels for UTI susceptibility and resistance (Figure 3).

Serum CRP level has a moderate predictive ability (AUC = 0.63 for susceptibility; AUC = 0.66 for resistance). Serum procalcitonin level has a poor predictive ability (AUC = 0.54 for susceptibility; AUC = 0.53 for resistance).

## 4. Discussion

Bladder cancer is a common malignancy seen mostly in men at an advanced age, as in our cohort [27,28]. Intravesical BCG therapy has been a mainstay in the treatment of non-muscle-invasive bladder carcinoma, owing to its well-documented efficacy in reducing recurrence and progression rates [7,8,9]. BCG causes an increased local immune response in the bladder which results in inflammation and anti-tumor activity [29]. However, the utility of this therapeutic approach is not without challenges; with adverse events and complications, particularly UTIs, warranting scrutiny.

UTIs associated with BCG therapy represent a notable clinical concern [28,30,31]. There are also several additional conditions, such as comorbidities affecting occurrence of UTIs. DM2 and CKD were significantly associated with both susceptible and resistant UTIs in our study. The link between DM2 and UTIs has been well documented, with hyperglycemia providing a favorable environment for bacterial proliferation and impairing immune function [32,33,34,35,36]. Similarly, CKD compromises immune defenses, increasing susceptibility to infections by opportunistic and resistant pathogens [37]. Our findings corroborate these previous reports, reinforcing the need for vigilant monitoring of diabetic and CKD patients undergoing BCG therapy. Yang et al. could not find a significant association between DM2, CKD and complications of BCG including UTIs [28]. Prabharasuth et al. investigated bladder cancer management in which the cancer had occurred after renal transplantation and could not detect any UTIs in patients who had taken intravesical BCG treatment [38]. Postmenopausal situation is a well-known cause of UTIs [34]. However, we could not find a significant correlation between postmenopausal situation and UTIs in patients treated with intravesical BCG.

Hematuria, which may be associated with traumatic urinary catheterization, is also considered as a common local toxicity/complication of intravesical BCG instillation [39,40,41]. In our study, hematuria was considered as a possible local toxicity and/or UTI symptom.

Pyuria is associated with poor outcomes for intravesical BCG treatment in the literature [42,43]. Lamm et al. and Di Lorenzo et al. considered pyuria as local toxicity [39,44]. In our study, we observed that pyuria may indicate UTIs. After the successful treatment of UTIs, pyuria diminished. This finding underlines the importance of urine cultures in case of pyuria.

Frequency, urgency, hesitancy and dysuria were the most common symptoms of UTIs in our study, which is compatible with the literature [31,39,41,45].

Recognizing these complaints and findings is crucial for healthcare providers when assessing patients for UTI-related complications during BCG therapy. While these symptoms and findings are not exclusive to UTIs, their presence should prompt further evaluation and diagnostic measures, considering the challenges in distinguishing UTIs from BCG-related effects.

Our study demonstrated that an increased number of BCG cycles significantly increased the likelihood of both susceptible and resistant UTIs. These findings align with previous studies indicating that prolonged intravesical BCG exposure may lead to bladder inflammation and microbial dysbiosis, thereby predisposing individuals to infections [46]. Additionally, an increased risk of urinary infection may have been observed due to urinary catheterization performed each time during the procedure. While our study found a significant association between BCG cycles and the development of UTI, other studies have suggested that the immunomodulatory effects of BCG may paradoxically enhance local immune defenses, reducing infection rates in some cohorts [47].

Our study reveals that sensitive bacterial pathogens precipitated UTIs in 25.8% of patients during the treatment course. Predominantly, these infections were attributed to *Escherichia coli*; this is consistent with previous findings [17,18,48]. However, other pathogens such as *Klebsiella pneumonia*, *Proteus mirabilis*, *Pseudomonas aeruginosa* and *Streptococcus agalactiae* were also identified, underscoring the diverse microbial landscape of BCG-related UTIs. Development of resistant UTIs could also be classified as nosocomial infections due to a urinary catheterization. The inclusion of additional variables, such as comorbidities, allowed us to further understand the factors contributing to UTI susceptibility.

A noteworthy observation was the emergence of resistant UTIs, afflicting 13.3% of patients during follow-up. Notably, within this subset, *E. coli*, *P. aeruginosa* and *K. pneumonia* were the predominant causative agents which can be nosocomial agents. The rate of UTIs caused by *P. aeruginosa* increased in this group, as expected.

Since our study focused solely on the first episode of urinary tract infection, patients did not receive antibiotic prophylaxis. However, close follow-up is essential for those with recurrent UTIs, and they should be evaluated for potential antibiotic prophylaxis in such cases. However, it should be noted that antibiotic prophylaxis may cause the development of more resistant infections. Recent studies have explored alternative strategies for UTI prevention in urological procedures. Quattrone et al. [49] demonstrated that D-Mannose plus *Saccharomyces boulardii* significantly reduced the incidence of UTIs and local discomfort following cystoscopy. While this study focused on cystoscopy-related infections, similar prophylactic approaches could be considered in patients undergoing IV BCG therapy, especially given the high UTI rates observed in our cohort. Future research should explore whether such interventions could mitigate BCG-associated UTI risks and improve patient outcomes.

We have observed that DM2 and CKD may influence the type of infecting pathogen. This is potentially due to compromising of immune defenses or immunological alterations making individuals more prone to infections with opportunistic or more resistant bacterial strains. BPH also has an effect on pathogen type of UTIs. The structural changes in the urinary tract due to BPH might create a favorable environment for more resistant or hospital-acquired pathogens.

Diagnosing UTIs in this context presents a unique clinical puzzle. Although patients manifested classic urinary symptoms and pyuria (not uncommon in individuals undergoing intravesical BCG therapy), fever was notably absent. We wondered if we could use acute phase reactants for distinguishing local toxicities and UTIs. While serum CRP levels were significantly elevated in UTI patients, particularly those with resistant infections, serum procalcitonin levels did not show significant differentiation between UTI types. This is consistent with the work of Schuetz et al., who found that serum procalcitonin level is a reliable marker for systemic infections but may not be as effective in distinguishing localized infections such as cystitis [50]. The weak correlation between BCG cycles and serum CRP/procalcitonin levels suggests that while BCG therapy contributes to inflammation, it does not consistently drive systemic inflammatory marker elevation. Our results, therefore, support the hypothesis that serum CRP levels can serve as an adjunct diagnostic tool for UTIs but still seems to be not sufficient for distinguishing between inflammation and cystitis properly. These observations again highlight the importance of meticulous clinical assessment and urine culture confirmation for accurate diagnosis.

It is imperative to acknowledge the potential for disseminated BCG infection, as reported before [51,52,53,54]; albeit it is rare, as highlighted in our study (0.4%). Such cases necessitate prompt recognition and aggressive management to avert potentially severe consequences. The documentation of immunosuppressive situations and traumatic catheterization may enable a more comprehensive assessment of patient risk factors for disseminated infection, aiding in early detection and intervention [55].

The findings of our study have significant clinical implications. Patients undergoing multiple BCG cycles should be closely monitored for UTI development, with prompt intervention to prevent progression to resistant infections. Diabetic and CKD patients require heightened surveillance, given their increased susceptibility to both susceptible and resistant UTIs. Serum CRP level may serve as a useful adjunctive biomarker for UTIs in this population, while serum procalcitonin level appears to have limited diagnostic value. Antimicrobial stewardship programs should prioritize screening for ESBL-producing pathogens, as they constitute a growing concern in UTI management.

Although our study provides valuable insights, certain limitations should be acknowledged, such as the single-center design, which may limit the generalizability of findings to broader populations.

## 5. Conclusions

This study highlights the intricate relationship between BCG therapy, comorbidities, inflammatory markers and UTI risk. Our findings are largely consistent with the existing literature, reinforcing known risk factors while identifying new insights into microbial trends and inflammatory marker utility. A better understanding of these interactions will aid in refining clinical management strategies and optimizing patient outcomes in BCG-treated populations.

## Figures and Tables

**Figure 1 medicina-61-00379-f001:**
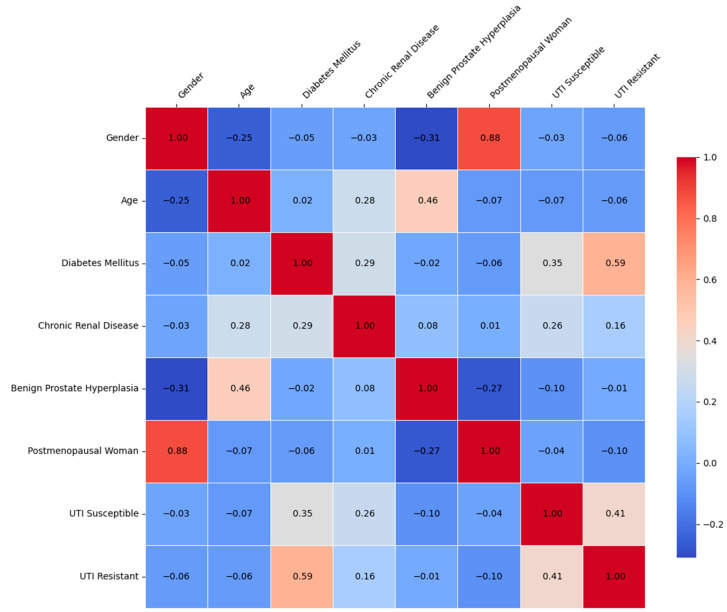
Correlation heatmap of demographic variables, comorbidities and UTIs.

**Figure 2 medicina-61-00379-f002:**
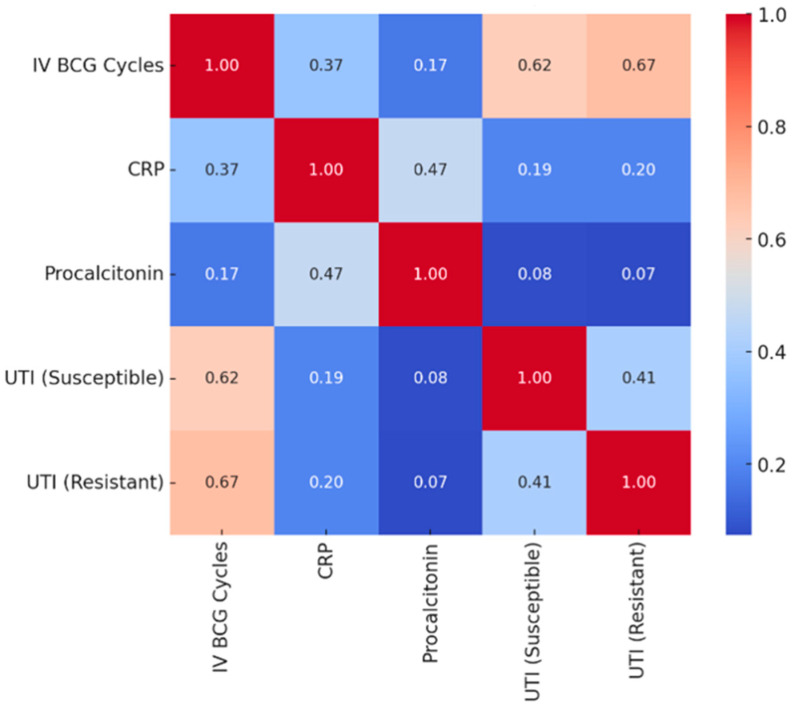
Correlation heatmap of intravesical BCG cycles, acute phase reactants and UTIs.

**Figure 3 medicina-61-00379-f003:**
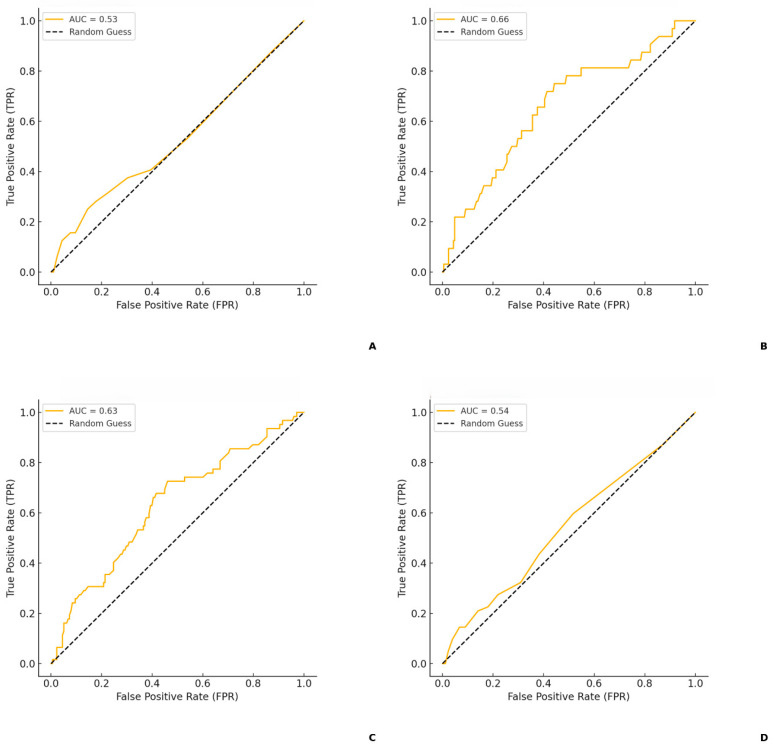
(**A**): ROC curve for serum procalcitonin levels (UTIs caused by resistant pathogens). (**B**): ROC curve for serum CRP levels (UTIs caused by resistant pathogens). (**C**): ROC curve for serum CRP levels (UTIs caused by susceptible pathogens). (**D**): ROC curve for procalcitonin levels (UTIs caused by susceptible pathogens.

**Table 1 medicina-61-00379-t001:** Comparison of the variables of UTI groups.

Parameters	S-UTI	R-UTI
Median age (IQR)	63 (14.75)	63 (10.5)
Male (%)	54 (87.09)	29 (90.6)
Female (%)	8 (12.91)	3 (9.4)
Postmenopausal female (%)	6 (9.6)	1 (3.1)
DM2(%)	20 (32.2)	20 (62.5)
CKD (%)	18 (29.03)	9 (28.1)
BPH (%)	17 (27.4)	11 (34.3)
Urgency (%)	33 (53.2)	20 (62.5)
Frequency (%)	24 (38.7)	16 (50)
Hesitancy (%)	31 (50)	15 (46.8)
Dysuria (%)	25 (40.3)	10 (31.2)
Hematuria (%)	21 (33.8)	8 (25)
Pyuria (%)	22 (35.4)	11 (34.3)
Median intravesical BCG Cycles (IQR)	8 (4)	10 (7)
Median serum CRP level (mg/L) (IQR)	28.8 (23.6)	31 (18.3)
Median serum procalcitonin level (ng/mL) (IQR)	0.2 (0.4)	0.2 (0.5)
Pathogens		
*Escherichia coli* (%)	43 (69.35)	14 (43.75)
*Escherichia coli* + *Pseudomonas aeruginosa* (%)	5 (8.06)	5 (15.62)
*Klebsiella pneumonia* (%)	4 (6.45)	5 (15.62)
*Proteus mirabilis* (%)	4 (6.45)	-
*Escherichia coli* + *Klebsiella pneumonia* (%)	2 (3.23)	2 (6.25)
*Pseudomonas aeruginosa* (%)	2 (3.23)	5 (15.62)
*Pseudomonas aeruginosa* + *Klebsiella pneumonia* (%)	1 (1.61)	1 (3.12)
*Streptococcus agalactiae* (%)	1 (1.61)	-

S-UTI: Urinary tract infections caused by susceptible pathogens; R-UTI: urinary tract infections caused by resistant pathogens; SD: standard deviation; DM2: diabetes mellitus type 2; CKD: chronic kidney disease; BPH: benign prostate hyperplasia; BCG: *Bacillus-Calmette-Guerin*; CRP: C-reactive protein.

**Table 2 medicina-61-00379-t002:** Multivariate logistic regression analysis of variables according to UTI types.

	Coefficient (β)		*p*-Value	
Variables	S-UTI	R-UTI	S-UTI	R-UTI
BCG cycles	0.48	0.91	0.00002	0.000001
CKD	−1.41	−0.52	0.01	0.123
Age	0.03	0.07	0.65	0.44
Serum CRP levels (mg/L)	0.10	0.15	0.321	0.298
Serum procalcitonin levels (ng/mL)	0.12	0.11	0.482	0.502
Gender	0.21	−0.32	0.539	0.371
DM2	0.43	−3.35	0.205	0.000002

S-UTI: urinary tract infections caused by susceptible agents; R-UTI: urinary tract infections caused by resistant agents; BCG: *Bacillus Calmette-Guerin*; CKD: chronic kidney disease; CRP: C-reactive protein; DM2: diabetes mellitus type 2.

**Table 3 medicina-61-00379-t003:** Comparison of acute phase reactants, BCG cycles according to UTI types.

Variable	Group	Mean Rank	U-Statistic	*p*-Value
Serum CRP level (mg/L)	S-UTI	50.2	300.1	0.001
	R-UTI	47.3	290.2	0.003
Serum procalcitonin Level (ng/mL)	S-UTI	48.5	315.4	0.059
	R-UTI	49.1	310.7	0.082
BCG cycles	S-UTI	60.3	200.4	0.00001
	R-UTI	59.8	205.7	0.00001

CRP: C-reactive protein; BCG: *Bacillus Calmette-Guerin*; S-UTI: Urinary tract infections caused by susceptible pathogens; R-UTI: Urinary tract infections caused by resistant pathogens.

**Table 4 medicina-61-00379-t004:** Association between comorbidities and UTI types.

Comorbidity	Group	Chi-Square (χ²)	*p*-Value
DM2	S-UTI	15.12	0.00009
	R-UTI	12.33	0.0004
CKD	S-UTI	9.88	0.001
	R-UTI	7.54	0.006
BPH	S-UTI	2.45	0.117
	R-UTI	1.88	0.17
Postmenopausal status	S-UTI	2.12	0.145
	R-UTI	1.75	0.187

DM2: diabetes mellitus type 2; CKD: chronic kidney disease; BPH: benign prostate hyperplasia; S-UTI: urinary tract infections caused by susceptible pathogens; R-UTI: urinary tract infections caused by resistant pathogens.

**Table 5 medicina-61-00379-t005:** Association between comorbidities and pathogen types.

Comorbidity	Chi-Square (X²)	*p*-Value	Degrees of Freedom (DoF)
DM2	42.007	0.000001	7
CKD	20.110	0.005	7
BPH	16.080	0.024	7
Postmenopausal situation	2.924	0.891	7

DM2: diabetes mellitus type 2; CKD: chronic kidney disease; BPH: benign prostate hyperplasia.

## Data Availability

Data are contained within the Appendix A.

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
