# Peer review of "Intravesical BCG: A Double-Edged Sword? The Untold Story of Infection Risks"

_medicina, 2025, doi:10.3390/medicina61030379_

Round 1
Reviewer 1 Report
Comments and Suggestions for Authors
This paper reported the correlation between intravesical instillation of BCG and UTIs. The reviewer would like to suggest some critiques to make this paper as follows.
- On line 9, ‘Intravesical (IV) BCG therapy’ is not wrong, but it is an expression that may induce misunderstandings in the reader. “IV” is usually taken to mean intravenous. The reviewer thinks that intravesical instillation of BCG is adequate.
- On line 19, “IV BCG cycle” is inadequate. “the number of~” is correct.
- On line 26, the authors should change from “UTI development” to “the risk of developing a UTI.”
- On line 41, “urothelial cancer” should be used instead of “urothelial carcinoma.”
- On line 46, this sentence is unclear. Many other intravesical agents? None of which have..?
- On line 60, “the perspectives … experts” is unclear. The expert of …?
- In the way the data is described, mean and SD seem inappropriate. Does this data follow a normal distribution? The data should be used median and IQR.
- The authors should also revise the way the p-values are listed in the Tables.
- Normally, the significant digits of the p-value should be standardized to three digits.
Reviewer 2 Report
Comments and Suggestions for Authors
The authors in their article describe with good vision the risk of infection-inflammation in BCG therapies. A truly real life focus on these patients.
Topic of current interest. Good general organization of the paper. Proper methodology. Sufficient use of English.
I have only minor comments to improve the paper:
1) It does not describe any prophylaxis
2) could be cited:
D-Mannose Plus Saccharomyces boulardii to Prevent Urinary Tract Infections and Discomfort after Cystoscopy: A Single-Center Prospective Randomized Pilot Study
Quattrone, C. Medicina (Lithuania), 2023, 59(6), 1165
Round 2
Reviewer 1 Report
Comments and Suggestions for Authors
none.